# Machine-Learning-Based Clinical Biomarker Using Cell-Free DNA for Hepatocellular Carcinoma (HCC)

**DOI:** 10.3390/cancers14092061

**Published:** 2022-04-20

**Authors:** Taehee Lee, Piper A. Rawding, Jiyoon Bu, Sunghee Hyun, Woosun Rou, Hongjae Jeon, Seokhyun Kim, Byungseok Lee, Luke J. Kubiatowicz, Dawon Kim, Seungpyo Hong, Hyuksoo Eun

**Affiliations:** 1Department of Biomedical Laboratory Science, Daegu Health College, Daegu 41453, Korea; taehee1155@gmail.com; 2Department of Senior Healthcare, Graduate School, Eulji University, Uijeongbu-si 11759, Korea; hyunsh@eulji.ac.kr; 3Pharmaceutical Sciences Division, School of Pharmacy, University of Wisconsin—Madison, Madison, WI 53705, USA; rawding@wisc.edu (P.A.R.); jbu@inha.ac.kr (J.B.); kubiatowicz@wisc.edu (L.J.K.); dkim634@wisc.edu (D.K.); 4Wisconsin Center for NanoBioSystems (WisCNano), University of Wisconsin—Madison, Madison, WI 53705, USA; 5Department of Biological Sciences and Bioengineering, Inha University, Incheon 22212, Korea; 6Industry-Academia Interactive R&E Center for Bioprocess Innovation, Inha University, Incheon 22212, Korea; 7Department of Internal Medicine, Chungnam National University Sejong Hospital (CNUSH), Sejong 30099, Korea; rws00@cnuh.co.kr (W.R.); kyniker@naver.com (H.J.); 8Department of Internal Medicine, Chungnam National University Hospital, Daejeon 35015, Korea; midoctor@cnu.ac.kr (S.K.); gie001@cnuh.co.kr (B.L.); 9Yonsei Frontier Lab, Department of Pharmacy, Yonsei University, Seoul 03722, Korea; 10Department of Internal Medicine, College of Medicine, Chungnam National University, Daejeon 35015, Korea

**Keywords:** cell-free DNA (cfDNA), circulating tumor DNA (ctDNA), hepatocellular carcinoma (HCC), liquid biopsy, principal component analysis (PCA)

## Abstract

**Simple Summary:**

Circulating cell-free DNA (cfDNA) has attracted a great deal of scientific interest as a predictive biomarker for the diagnosis and prognosis of hepatocellular carcinoma (HCC). HCC result in high mortality due to the absence of blood biomarkers for early diagnosis and prognosis. We established cfD_HCC_ as a new scoring system by applying a machine learning algorithm that integrates the expression profiles of cfDNA. Based on this, it was possible to accurately predict the clinico-pathological characteristics of patients with HCC as well as improve their survival.

**Abstract:**

(1) Background: Hepatocellular carcinoma (HCC) is one of the leading causes of cancer-related death worldwide. Although various serum enzymes have been utilized for the diagnosis and prognosis of HCC, the currently available biomarkers lack the sensitivity needed to detect HCC at early stages and accurately predict treatment responses. (2) Methods: We utilized our highly sensitive cell-free DNA (cfDNA) detection system, in combination with a machine learning algorithm, to provide a platform for improved diagnosis and prognosis of HCC. (3) Results: cfDNA, specifically alpha-fetoprotein (AFP) expression in captured cfDNA, demonstrated the highest accuracy for diagnosing malignancies among the serum/plasma biomarkers used in this study, including AFP, aspartate aminotransferase, alanine aminotransferase, albumin, alkaline phosphatase, and bilirubin. The diagnostic/prognostic capability of cfDNA was further improved by establishing a cfDNA score (cfD_HCC_), which integrated the total plasma cfDNA levels and cfAFP-DNA expression into a single score using machine learning algorithms. (4) Conclusion: The cfD_HCC_ score demonstrated significantly improved accuracy in determining the pathological features of HCC and predicting patients’ survival outcomes compared to the other biomarkers. The results presented herein reveal that our cfDNA capture/analysis platform is a promising approach to effectively utilize cfDNA as a biomarker for the diagnosis and prognosis of HCC.

## 1. Introduction

Liver cancer is a significant global health challenge and is the fourth leading cause of cancer-related deaths worldwide with an estimated incidence of more than 1 million cases by 2025 [1,2]. It is a remarkably heterogeneous cancer type and has multiple properties, with variations in etiology, inter-, and intra-phenotypes [3]. Hepatocellular carcinoma (HCC) is the most common form of liver cancer and accounts for ~90% of all cases [4]. Recent studies have shown that the molecular pathogenesis of HCC is a dynamic process that involves the progressive accumulation of several abnormalities and alterations, including mutations and chromosomal aberrations that drive hepatocarcinogenesis [5]. Although less than 10% of the abnormalities are mutations, dominant mutational drivers in HCC remain undruggable [1]. Furthermore, several risk factors have been identified, including chronic liver disease, cirrhosis, infection with hepatitis B virus (HBV) or C virus (HCV) infection, alcohol abuse, nonalcoholic fatty liver disease (NAFLD), obesity, diabetes mellitus, tobacco, and diet [6,7,8]. Surgical intervention, including resection and liver transplantation, is currently the most effective therapy for HCC patients, with a 5-year survival incidence of 70–80%. However, most HCC is detected at advanced stages, and only 20–30% of patients are eligible for surgical intervention [1,9]. HCC prognosis is based on the tumor stage, with a progressive decrease in survival from 70% at 5 years for early stage HCC with therapeutic intervention, compared to a median survival of ~1–1.5 years for advanced-stage HCC [1]. Therefore, it is critical to identify HCC at early stages to enable the use of curative treatments and improve overall survival outcomes [2].

Tissue biopsies are the current gold standard for diagnosing HCC; however, they provide a limited snapshot of tumors and vary in sensitivity due to difficulties in identifying lesions, distinguishing HCC from dysplastic nodules, are difficult to obtain, and histologic diagnosis for small nodules (≤1 cm) is challenging [1,10,11]. HCC is one of the few types of tumors that can be diagnosed based on non-invasive criteria, including computed tomography (CT) scans, ultrasonography (US) screening, and dynamic-contrast-enhanced magnetic resonance imaging (MRI) [12]. Nevertheless, these techniques often result in inconclusive or false-negative diagnoses [10,13]. Therefore, current HCC diagnosis guidelines recommend regular biannual US screening along with serum α-fetoprotein (AFP) screening for patients who are considered to be at risk for HCC [14]. AFP is a biomarker for HCC, and high AFP levels (>10 ng/dL) can be indicative of underlying malignant pathology [15]. However, AFP alone has been shown to lack adequate sensitivity and the specificity required for effective surveillance and diagnosis of HCC. For instance, serum AFP levels are elevated in HCC patients and in those with liver diseases, including cirrhosis, chronic liver failure, and hepatitis [16]. Other serum liver enzymes such as aspartate aminotransferase (AST), alanine aminotransferase (ALT), albumin, alkaline phosphatase (ALP), and bilirubin have been investigated as potential biomarkers for HCC; however, the diagnostic and prognostic capabilities of these serum biomarkers remain unclear. 

Currently, a plethora of candidate biomarkers, such as cell-free DNA (cfDNA), exosomes, and circulating tumor cells (CTCs), are being investigated as biomarkers for the diagnosis and prognosis of different cancers, including HCC [17,18,19,20,21,22,23,24,25,26,27]. Among these biomarkers, cell-free DNA (cfDNA) has garnered a great deal of scientific and clinical interest as an indicator of tumor progression and predictor of therapeutic responses due to its potential significance in determining the molecular properties and genetic alterations in tumors [28]. Specifically, several studies have affirmed that serum cfDNA levels and the expression of specific oncogene are highly associated with poor prognosis and the potential for tumor recurrence [29,30]. However, the utilization of cfDNA as a biomarker has not yet been realized in the clinic due to the relatively large amount of blood (1–10 mL) required for cfDNA detection and the inherent difficulties of cfDNA isolation, collection, and processing [19]. Therefore, there is an increasing demand for advanced technology that can detect and capture cfDNA using a small amount of blood. Previously, we devised a method using polydopamine–silica hybrids (PDA-SiO_2_) that increases the detection sensitivity of cfDNA by incorporating the molecular interaction between PDA nucleobases and silica–phosphate backbones [29,31]. 

Here, using our novel cfDNA capture system, we aimed to investigate the diagnostic and prognostic potential of cfDNA as a biomarker for HCC (Figure 1A). To validate the diagnostic capability of cfDNA, we quantified the amount of plasma cfDNA and *AFP* expression (cfAFP-DNA)in the captured cfDNA from several cohorts, including patients with HCC, alcoholic liver hepatitis (LA), or liver cirrhosis (LC). In addition, we examined the capability of levels of cfDNA to determine the pathological status of a tumor and estimate the survival outcomes by comparing cfDNA with the concentrations of serum liver enzymes, including total protein, AFP, AST, ALT, albumin, ALP, and bilirubin. To provide more reliable clinical information using our system, we used a machine learning technique [9,31,32,33,34] for our bead-based liquid biopsy assay. Specifically, we stratified HCC patients into subgroups based on the plasma cfDNA levels and cfAFP-DNA expression using *k*-means clustering and established an integrated cfDNA score for HCC patients (cfD_HCC_) using principal component analysis (PCA). We validated the diagnostic and prognostic capability of the integrated cfD_HCC_ score. The results showed that cfDNA is a promising biomarker for HCC, and the utilization of our highly sensitive cfDNA capture system, in combination with a machine learning algorithm, can potentially provide an improved platform for the diagnosis and prognosis of HCC.

## 2. Materials and Methods

### 2.1. Materials

Sodium alginate, calcium chloride (CaCl_2_), silica solution (LUDOX1 AM colloidal silica, 30 wt. % in H2O), 2-(3,4-Dihydroxyphenyl)ethylamine hydrochloride (dopamine hydrochloride), 1-(3-Dimethylaminopropyl)-3-EthylcarbodiimideHydrochloride (EDC), and N-Hydroxy- Succinimide (NHS) were all obtained from Sigma–Aldrich (St. Louis, MO, USA). The blood samples and clinical data were provided by the National University Hospital biobank of Gyeongsang, Chungnam, Kangwon, Jeonbuk, and Chungbuk, a member of the Korea Biobank Network. Proteinase K, AW1 wash buffer, SYBR Green Master Mix (2× Rotor-Gene SYBR Green PCR Master Mix), and nuclease-free water were purchased from Qiagen Inc. (Valencia, CA, USA). Ethyl alcohol was purchased from Samchun Inc. (Pyeongtaek, Gyeonggi-do, Korea). A High Sensitivity DNA Kit was purchased from Agilent Technologies (Santa Clara, CA, USA). All other chemicals used in this study were obtained from Sigma-Aldrich and used without further purification unless otherwise noted.

### 2.2. Human Study Design and Population

This retrospective cohort study was conducted in Chungnam National University Hospital in South Korea. Participants were divided into four groups based on their underlying clinicopathology: 152 patients diagnosed with HCC, 43 patients with LC, 24 patients with LA, and 30 healthy donors (HD). Note that HCC was diagnosed either by histopathological analysis of the biopsy specimen and/or radiological evaluations by medical doctors. A total of 91 patients (60%) were biopsy-proven. The present study was approved through an institutional review board (IRB) of Chungnam National University Hospital (CNUH-2019–04-052), and all participants completed an informed consent process before any experiments. 

### 2.3. Human Data Collection

The blood sample and clinical information were provided by the National University Hospital biobank of Kangwon and Chungbuk, and Chungnam, a member of the Korea Biobank Network. The Biobank Study’s ethical approval was previously granted by the IRB. The biobank sample was also defined using the codes for International Classification of Diseases (ICD). 

The basic demographic data collected included the participants’ age and sex. Additional health check-up data included height (cm), weight (kg), body mass index (BMI) (kg/m^2^), and blood pressure (mmHg) measurements. The participants were also provided with a self-report questionnaire during the check-up, asking about the presence of diabetes (yes or no), alcohol status (drinks or does not drink), smoking status (current smoker or does not smoke), and presence of HBV (positive or negative). During the check-up, 5 mL of whole blood was drawn from each participant. Approximately 200 µL of plasma was processed and stored accordingly. Clinical information, including histopathological stage, histology type, cell differentiation, cancer-associated marker expression, and serum antigen levels, were determined based on the physical exam, computed tomography (CT) scan, biopsy, immunohistochemistry (IHC), and/or blood tests from the blood bank. 

### 2.4. PDA-SiO_2_ Beads’ Preparation for cfDNA Capture

A sodium alginate solution (5% *w*/*v*) was dropped into a 100 mM CaCl_2_ aqueous solution and incubated for 1 h at room temperature to form alginate beads. After incubation, the beads were gently washed with deionized water (DW) three times and stored in DW before use. The carboxylic groups on the surface of the alginate beads were activated with 5 mM of EDC/NHS for 1 h at room temperature and then reacted with 5 mM of dopamine hydrochloride for 12 h. The pH of the dopamine hydrochloride solution was controlled by adding 0.1% hydrochloric acid (HCl) to maintain pH of ~7.0 using Tris–HCl buffer. The beads were then washed with DW and incubated in 1 mL of silica solution for 1 h before being used for DNA analysis. 

### 2.5. cfDNA Capture

Blood plasma was collected from 3 to 5 mL human blood samples after blood aliquot was centrifuged at 2990 rpm for 10 min. The plasma was pre-treated with proteinase K in a 10:1 volume ratio (200 μL: 20 μL) for 3 h at 37 °C. The samples were mixed with 200 μL of lysis buffer and incubated for 10 min at 37 °C, followed by additional mixing with 200 μL of 95% ethyl alcohol. The PDA–silica-coated beads were then dipped into the pre-treated samples with 20 μL of CaCl_2_ solution. The DNA fragments were reacted with the PDA–silica hybrids for 10 min under gentle agitation. The beads were then washed with 350 µL of AW1 buffer (Qiagen, Hilden, Germany) and stored in 50 µL of RNase/DNase free water. The plasma cfDNA fragments were measured using the Agilent Bioanalyzer 2100 instrument and High Sensitivity DNA Kit, a microfluidics-based platform, to determine the mean size for each defined smear region of plasma cfDNA and display electropherogram for each sample. The concentration of cfDNA was determined to range between 100 and 500 bp (Appendix A). The details can be found in our previous publications [29,31]. 

### 2.6. Real-Time Quantitative Polymerase Chain Reaction (qPCR)

qPCR (RG6200, Corbett Research, Sydney, Australia) was performed in triplicate using SYBR Green Master Mix (2× Rotor-Gene SYBR Green PCR Master Mix, Qiagen) at a final volume of 25 µL. The PCR cycling condition was as follows: 95 °C for 5 min with 40 cycles, 95 °C for 5 s, and 60 °C for 10 s. cfAFP-DNA was obtained by comparing *AFP* with the reference gene, RRP30 (2^−∆Ct^). Ct values above 40 were excluded from the data analysis. The primer sequences utilized included: an AFP forward primer (5′- AAA TGC GTT TCT CGT TGC -3′), AFP reverse primer (5′- GCC ACA CGG CCA ATA GTT TGT -3′), RPP30 forward primer (5′- GAT TTG GAC CTG CGA GCG 3′), and RPP30 reverse primer (5′ GCG GCT GTC TCC ACA AGT -3′). 

Droplet digital polymerase chain reaction (ddPCR; QX200, Bio-Rad, Hercules, CA, USA) was used in this study (for validation of AFP signal from the cell culture supernatant), and each assay mixture was performed in 20 µL reaction volumes. The mixture consisted of up to 30 ng of extracted DNA (1 µL), 2X EvaGreen ddPCR Supermix (10 µL), all individual primers (1 µL), and distilled water (7 µL). The ddPCR assay mixture was loaded into a disposable droplet generator cartridge, and 70 µL of droplet generation oil was loaded into each of the eight oil wells. The cartridge was then placed inside the QX200 droplet generator, and the droplets were transferred to a 96-well PCR plate. The plate was heat-sealed with foil and placed in a conventional thermal cycler (Bio-Rad, T100). We partitioned each reaction mixture into approximately 12,000–20,000 droplets with a droplet generator and then cycled them under the following conditions: 95 °C for 5 min (1 cycle); 95 °C for 30 s, and 55 °C for 1 min (40 cycles); 4 °C for 5 min; 90 °C for 5 min (1 cycle); and 4 °C hold. Cycled droplets were read individually with the QX200 droplet-reader (Bio-Rad). 

### 2.7. Statistical Analysis

The concentrations of peripheral blood biomarkers, including cfDNA, AFP, AST, ALT, total protein (TP), albumin, platelet, ALP, and total bilirubin (TB) were compared based on a Student’s *t*-test or Mann–Whitney U test, depending on the normality of the sample distribution. The clinical capabilities of these biomarkers were further assessed by constructing a receiver operating characteristic (ROC) curve. The prognostic value of cfDNA was validated using Kaplan–Meier plots and a Cox regression model for overall survival (OS) and disease-free survival. A value of *p* < 0.05 was regarded as statistically significant. All statistical analyses were carried out with SPSS Statistics 26 (SPSS, Chicago, IL, USA). 

## 3. Results and Discussion

### 3.1. Capture of cfDNA and AFP DNA for Analysis Using a New Bead-Based System

In a previous study, we developed a highly sensitive cfDNA capture system using PDA-SiO_2_-coated beads and demonstrated its diagnostic accuracy and clinical potential [29,31]. PDA catechol groups effectively adsorb DNA bases through metal coordination in the presence of polyvalent metal ions, particularly Ca^2+^. Additional π-π stacking and hydrogen bonding may further enhance DNA adsorption [35]. We applied this system for the diagnosis and prognosis of HCC by assessing the *AFP* expression from the captured cfDNA (cfAFP-DNA). The detection of cfAFP-DNA was also assessed using plasma samples obtained from three HCC patients. Our bead-based assay captured 1.69-fold (*p* = 0.123) more cfDNA copies than commercial kits (Figure 1B). Although the difference relative to commercial kits was not statistically significant at the 5% level, more cfAFP-DNA copies were detected from samples processed with PDA-SiO_2_-coated beads compared to those from the commercially available QIAamp DNA mini-kits for all three patients’ samples. Taken together, the results from this study and our previously published analyses support the high cfDNA detection sensitivity of our bead-based assay [29,31].

### 3.2. Baseline Clinical Characteristics

The baseline characteristics of 249 patients, of which 152, 43, 24, and 30 patients were diagnosed with HCC, LC, LA, and HD, respectively, were determined based on their underlying clinicopathology at the time of check-up (Appendix A). Of the 152 patients with HCC, 43 had a high risk of poor prognosis based on their tumor’s multifocality, size, and location, and were treated with trans-arterial chemoembolization (TACE) before the blood draw (TACE-treated group). Among 109 non-TACE groups at the point of blood draw, 27 patients were treated with TACE after the blood draw based on the decision during their follow-up visits. The proportion of males was higher in patients with HCC, LC, and LA (79.0%, 76.7%, and 79.2%, respectively) and lower in HDs (3.33%). HBV infection was also found in 57.9% (88/152) and 16.3% (7/43) of patients with HCC and LC, respectively. A small but statistically significant difference in age between patients with HCC and the remaining three groups (5–10 years older; *p* < 0.030) was identified. Regarding the clinical variables, alcohol use, hypertension, diabetes, and smoking were more prevalent in the cancer cohort than in non-cancer cohorts; however, no significant difference was found in BMI among all four groups. 

### 3.3. cfDNA as a Potential Biomarker for the Diagnosis of HCC Tumor

Standard serum tests for liver function in current clinical settings include those for total protein, AFP, AST, ALT, albumin, ALP, and bilirubin. These tests can provide specific indicators for liver disease and guide diagnostics, help estimate severity, assess prognosis, and evaluate therapy [36,37]. Therefore, a tumor profile was created for each participant with measurements including concentration values for these factors (Figure 2A and Appendix A). The concentrations of serum AFP, AST, and ALT were significantly higher among patients with cancer than in the non-cancer cohorts (*p* < 0.001). However, serum AST and ALT levels were elevated among patients with HCC, LC, and LA. Therefore, the difference in concentrations of these serum enzymes was significant only when comparing them between patients with HCC and healthy donors. Serum AFP was the only serum enzyme that exhibited a statistically significant difference in a comparison between patients with cancer and the three non-cancer cohorts. Notably, the difference in concentration of serum proteins, albumin, ALP, and bilirubin was not statistically significant when comparing serum enzyme levels between the patients with HCC and non-cancer cohorts, and a higher concentration of the enzymes was observed among specific non-cancer cohorts relative to patients with HCC. As a result, AFP was the only serum enzyme marker that was elevated among patients with HCC. The area under the curve of the receiving operating characteristic (AUC-ROC) was calculated to be 0.836 (*p* < 0.001), 0.706 (*p* < 0.001), and 0.840 (*p* < 0.001) for AFP to discriminate between patients with HCC from patients with LC, LA, and healthy donors, respectively (Figure 2B,C and Appendix A). Similarly, we validated the potential of cfDNA as a diagnostic biomarker for HCC. As demonstrated in Figure 2A, plasma cfDNA levels were the highest among patients with HCC (median: 0.25 ng/µL), followed by LC (0.18 ng/µL), LA (0.11 ng/µL), and HDs (0.06 ng/µL), which had a log-rank *p*-value of <0.001. As a result, the plasma cfDNA demonstrated a significant diagnostic performance with an AUC-ROC of 0.713 (*p* = 0.001), 0.592 (*p* = 0.066), and 0.805 (*p* < 0.001) for differentiating HCC in the LC, LA, and HD groups, respectively. The diagnostic capability of plasma cfDNA was superior to other serum enzymes except for serum AFP. Therefore, we assessed *AFP* expression (cfAFP-DNA) from the captured cfDNA using qPCR. Interestingly, cfAFP-DNA expression demonstrated a better diagnostic capability than both plasma cfDNA levels and serum *AFP* expressions. The AUC-ROCs of cfAFP-DNA to differentiate HCC from LC, LA, and HDs were 0.861 (*p* < 0.001), 0.744 (*p* < 0.001), and 0.971 (*p* < 0.001), respectively, and these values were greater than those of plasma cfDNA and serum AFP (Figure 2B,C). 

To further explore the clinical utility of cfDNA in diagnosing HCC in comparison to conventional serum biomarkers, we calculated the sensitivity, specificity, positive predictive value (PPV), and negative predictive value (NPV) at the threshold, and gave the highest accuracy with a specificity of ≥0.7 for each biomarker (Figure 2D and Appendix A). Notably, cfAFP-DNA outperformed other serum biomarkers and plasma cfDNA for diagnosing HCC. More specifically, the accuracy of cfAFP-DNA for detecting HCC vs. LC, LA, and HDs was 0.738, 0.790, and 0.951, respectively, which was the highest amongst the biomarkers tested in this study. These results demonstrate the diagnostic potential of cfDNA, specifically cfAFP-DNA, as a reliable biomarker for detecting HCC.

### 3.4. cfDNA as a Potential Biomarker for Determining the Pathological Features of HCC Tumors 

We investigated the capability of cfDNA to estimate the pathological features of HCC, such as its stage, lymphovascular invasion (LVI), size, and multifocality (Appendix A). cfDNA was the only biomarker that could successfully stratify patients according to their modified International Union Against Cancer (UICC) stage, namely, the UICC stage ≥ II vs. stage < II (*p* = 0.009) and UICC stage ≥ III vs. stage < III (*p* < 0.001). A high plasma cfDNA concentration was also strongly associated with the presence of LVI (*p* = 0.017), along with high concentrations of serum AST (*p* = 0.018) and ALT (*p* = 0.002) (Appendix A). Furthermore, cfDNA was a good indicator for estimating the number of tumors present; the patients with multifocal/multicentric HCCs exhibited significantly higher cfDNA concentrations than those with solitary tumors (*p* = 0.042). Among the enzyme markers evaluated, serum ALP was the only marker that was associated with the number of tumors present. Further, the amount of cfDNA present in the plasma was strongly correlated with the size of the largest tumor present in the liver; the patients with tumor sizes greater than the median exhibited a higher concentration of plasma cfDNA than those with small tumors (*p* = 0.034). Serum AFP (*p* = 0.033), AST (*p* = 0.008), and ALT (*p* = 0.009) concentrations were also found to be highly associated with tumor size. High cfAFP-DNA expression tended to be associated with a high UICC stage, the presence of LVI, a large tumor size, and the multifocality of a tumor; however, the results were not statistically significant. 

Although the quantitative cfDNA analysis provided statistically significant results compared to serum enzymes for identifying individuals at high-risk for HCC, its accuracy was insufficient for application in clinical practice. Therefore, to improve the capability of cfDNA to determine the pathological status of HCC, we integrated the expression profiles of plasma cfDNA and cfAFP-DNA to establish a cfDNA score specific to HCC patients (cfD_HCC_) using a series of machine learning techniques, including *k*-means cluster analysis, the elbow method, and principal component analysis (PCA) (Appendix A). Patients were clustered into subgroups based on *k*-means clustering analysis, depending on their plasma cfDNA concentrations and *cfAFP-DNA* copy numbers. The cfD_HCC_ score was established by integrating the expression profiles of plasma cfDNA and *cfAFP-DNA* using PCA. The clinical performance of cfD_HCC_ for determining the higher UICC stages, the existence of LVI, large tumor sizes, and the multifocality of tumors was investigated and compared with other biomarkers used in this study. The ROC curve analysis indicated that the cfD_HCC_ score can differentiate patients with HCC based on their tumor UICC stage, detect multifocal tumors, and estimate tumor sizes with greater accuracy than either plasma cfDNA or cfAFP-DNA alone. Although the AUC-ROC value of the cfD_HCC_ score was not more accurate in predicting LVI in comparison to plasma cfDNA, the difference was not statistically significant (Figure 3A and Appendix A). A few serum enzymes exhibited high AUC-ROC values for detecting specific high-risk tumors, but overall, these enzyme markers demonstrated a low performance in comparison to the cfD_HCC_ score. For instance, serum ALP exhibited the highest AUC-ROC value (0.696) for detecting multifocal tumors among the biomarkers used in this study; however, ALP had an AUC-ROC < 0.5 for differentiating patients with late-stage HCC, detecting LVI, and estimating tumor sizes. 

The performance of each biomarker for differentiating high-risk tumors was further validated by setting the median value of each biomarker as the threshold (Figure 3B and Appendix A). The cfD_HCC_ score showed the highest accuracy among all the biomarkers of interest in differentiating UICC stages and estimating tumor size. The cfD_HCC_ score also demonstrated greater accuracy than the expression profiles of the total cfDNA and the relative amount of cfAFP-DNA for detecting LVI and multifocality. While serum AST, ALT, and ALP levels were more accurate than the cfD_HCC_ score for detecting LVI and multifocality, these markers demonstrated poor performance in detecting other phenotypes of HCC. These findings demonstrate the clinical potential of the combination of our new cfDNA detection system and machine-learning-based analysis method and can be used to predict the survival outcomes and estimate cancer recurrence.

We investigated cfDNA as a potential biomarker for pathological features from the original tumor. In 90/152 (59.2%) patients with HCC, we were able to obtain clinic-pathological information from the primary tumor. The results were examined further by analyzing the correlation between the levels of cfAFP DNA and the extent of the tumor burden. As shown in Appendix A, the levels of cfAFP DNA and serum AFP showed a positive correlation with the size of the tumor burden, with a Pearson’s correlation coefficients of 0.191 (*p* < 0.001) and 0.060 (*p* = 0.020). In contrast, the results obtained using cfDNA showed correlation coefficients below 0.004 (*p* = 0.541), indicating that none of these tests were representative of the tumor burden. Our study is based on modified UICC staging [38], which is a system that evaluates the stage of HCC based on imaging data, the number and size of the tumor, whether it has invaded blood vessels or bile ducts, and whether it has metastasized to the lymph nodes or distant organs. We used the modified UICC staging system because most patients with HCC had liver cirrhosis, and liver biopsy is often not easy to perform due to the risk of bleeding and tumor dissemination. Therefore, in this study, our results showed a correlation between pathological features and molecular data by limiting the analysis to a few patients with early-stage HCC who have undergone surgical treatment. The cfDNA AFP and serum AFP were reflected in the tumor size, as seen in Appendix A. Tumor size showed a diverse positive correlation coefficient. Unfortunately, in the Edmondson and cirrhosis grades, the correlation and significant *p*-value of the original tumor could not be evaluated. Since hepatic fibrosis or liver cirrhosis was diagnosed clinically based on imaging characteristics, blood test results, fibroscan, computed tomography, or magnetic resonance imaging in non-cancer patients, it is hard to identify a correlation between pathological characteristics and molecular data. To conclude, because the staging of patients with HCC was based on modified UICC staging using radiologic T staging, the correlation between molecular data for cfDNA and pathological information could be identified only for patients who received surgical treatment.

### 3.5. cfDNA as a Potential Biomarker for Predicting Survival Outcomes of Patients with HCC

To validate the prognostic potential of the cfD_HCC_ score, we stratified the HCC study subjects into two subgroups according to the TACE treatment status. TACE is considered a pre-treatment option for patients with intermediate HCC (BCLC stage B), who have multifocal tumors that are difficult to treat directly with hepatectomy and ablation therapy [39]. In our study, out of 152 HCC patients, 43 were pre-treated with TACE before their blood draw. Among these patients, we found that the TACE-pre-treated patients had a higher multifocality (~36%) than the non-TACE group (~17%). As multifocal tumors exhibit higher metastatic potential, heterogeneity, and proliferation rates than unifocal tumors, patients with multifocal tumors are reported to show a worse prognosis [40]. Due to this high multifocality, our study also demonstrated a higher overall death rate for the TACE-treated group (~26%) than the non-TACE group (~7%). Likewise, recurrence of HCC was detected in ~81% of TACE-treated patients and only in ~49% of non-TACE group patients. These results showed that the TACE-treated group was more susceptible to poor outcomes in comparison to the non-TACE group. 

Due to the high risk of recurrence and death, the predictive capability of the cfDNA detection and capture system was analyzed among the TACE-treated patients. A higher cfD_HCC_ score was associated with a poor prognosis in recurrence-free survival (RFS) analysis of TACE-treated patients (Figure 4A and Appendix A). The mean RFS among TACE-treated patients, presented as lower vs. higher than the median cfD_HCC_ score was 26.5 ± 5.2 vs. 15.9 ± 4.0 months (*p* = 0.061). Compared to the cfD_HCC_ score, the differences between the < median, and in the ≥ median groups was less significant for the other biomarkers used in this study, including cfDNA (26.0 ± 5.2 vs. 16.5 ± 4.0 months; *p* = 0.153). The cfD_HCC_ score could predict the occurrence of malignant recurrence, including the development of multifocal and marginal HCC recurrences (Figure 4B,C). For instance, TACE-treated patients with lower- vs. higher-than-the-median cfD_HCC_ score had RFS with multifocal and marginal recurrences of 54.0 ± 4.6 vs. 23.2 ± 5.5 months (*p* = 0.001) and 41.1 ± 6.3 vs. 27.4 ± 6.3 months (*p* = 0.061), respectively, which outperformed the other biomarkers used in this study (Appendix A). Likewise, the cfD_HCC_ score was superior to other biomarkers for predicting the overall survival (OS) of TACE-treated patients with a mean OS of 65.7 ± 4.4 vs. 47.7 ± 5.6 months (*p* = 0.077) between the groups having higher- and lower-than-the-median cfD_HCC_ scores (Figure 4D and Appendix A).

Univariate Cox regression analysis further demonstrated the prognostic capability of the cfD_HCC_ score (Figure 4E and Appendix A). The hazard ratios (HR) of the cfD_HCC_ scores (non-categorical model) were 1.569 (*p* = 0.001), 1.477 (*p* = 0.066), and 2.443 (*p* < 0.001) for predicting the recurrence, marginal recurrence, and survival, respectively, which were superior to the other biomarkers used in this study. The cfD_HCC_ score was also more capable of predicting multifocal recurrence in comparison to the other biomarkers (HR = 1.728; *p* = 0.005) except for total protein (HR = 1.779; *p* = 0.171), which had a higher HR but less significance. By setting the median as a cutoff, the binary model of the cfD_HCC_ score demonstrated the highest predictive capability with the highest statistical significance for among the four survival analyses with HRs of 1.875 (*p* = 0.073), 2.471 (*p* = 0.076), 8.544 (*p* = 0.005), and 3.123 (*p* = 0.093) for predicting the recurrence, marginal recurrence, multifocal recurrence, and survival, respectively. 

Notably, the prognostic capability of cfD_HCC_ score was modestly decreased when we included 27 patients who received TACE treatment during their follow-up period (after the blood draw) for the analysis. However, it still had a stronger correlation with OS and RFS than most of the other biomarkers used in this study. Interestingly, including the cfD_HCC_ score, none of the biomarkers used in this study demonstrated a clinical significance for estimating the survival of the non-TACE group, which had a low risk of malignancies and poor outcomes (Figure 4F and Appendix A). For estimating OS and RFS, the patients with a high cfD_HCC_ score still tended to have a comparably poor prognosis, although the results are statistically insignificant. However, considering that a comparably small subset of the non-TACE group died or had HCC recurrence compared to the TACE-treated group, we hypothesize that there exists potential clinical relevance between survival outcomes and the expression profiles of different biomarkers. This hypothesis will be tested in future studies, including a longitudinal study for evaluating various HCC biomarkers, clinical parameters, and prognostic outcomes with larger patient cohorts.

## 4. Conclusions

A significant challenge in HCC treatment is the lack of biomarkers that enable clinically reliable diagnosis and prognosis of HCC. Conventional tissue biopsy underrepresents tumor heterogeneity because the technique provides a snapshot of a small tumor fragment at a specific time point alone. Alternatively, serum enzymes have offered advantages in monitoring tumor heterogeneity; however, serum enzymes typically lack the adequate sensitivity and specificity needed to be utilized as a biomarker for the effective surveillance and diagnosis of HCC. To overcome these challenges, cfDNA has emerged as a novel biomarker for HCC. Liquid biopsy based on cfDNA analysis can detect actionable mutations or other molecular alterations, monitor treatment responses in real-time, and guide drug selection and dosing for HCC with high sensitivity [30]. Specifically, the copy numbers or expression levels of various HCC-associated oncogenes and suppressors, such as TP53, TERT, and ARID1A, have been assessed for the early detection and prognosis monitoring of HCC [41,42,43,44]. In this study, we combined our high-sensitive cfDNA detection assay and a machine-learning-based algorithm to further improve the diagnosis and monitoring of HCC. Built on our previously developed PDA-SiO_2_ hybrids for cfDNA detection for gastric tumors, we applied the detection and capture system in combination with machine learning to enable an accurate diagnosis and prognosis for HCC. By assessing *AFP* expression from captured cfDNA (cfAFP-DNA), we found that both total plasma cfDNA levels and cfAFP-DNA expression were elevated among patients with HCC. Specifically, cfAFP-DNA expression was superior at discriminating HCC patients from LC, LA, and HD cohorts compared to other serum enzymes, including AFP, AST, ALT, albumin, ALP, and bilirubin. The performance of our cfDNA detection system was further improved by utilizing a machine learning algorithm for the clinical analysis of HCC, which integrated the expression profiles of plasma cfDNA and the amount of cfAFP-DNA to establish a score specific for HCC patients (cfD_HCC_). cfD_HCC_ demonstrated improved accuracy in determining patients’ UICC stages, detecting multifocality and LVI, and estimating tumor sizes compared to either plasma cfDNA or cfAFP-DNA alone. Furthermore, cfD_HCC_ could predict HCC recurrence and survival outcomes more accurately than individual biomarkers used in this study. The diagnostic and prognostic capabilities of our new system can be further improved by leveraging additional HCC-associated biomarkers. Specifically, we are planning to analyze different types of HCC-associated genes (i.e., *TP53*, *TERT*, and *ARID1A*) from cfDNA and conduct a larger set of cluster analyses to further improve the clinical performance of our system. We will assess the diagnostic and prognostic capability of the biomarkers after the administration of various therapeutic options (i.e., radiofrequency ablation, chemotherapy, immunotherapy) for HCC patients. Taken together, our system revealed high diagnostic and prognostic capabilities and can be potentially utilized in the clinic as a reliable system for identifying HCC in early stages, guiding therapeutic decisions, and improving overall survival outcomes.

## Figures and Tables

**Figure 1 cancers-14-02061-f001:**
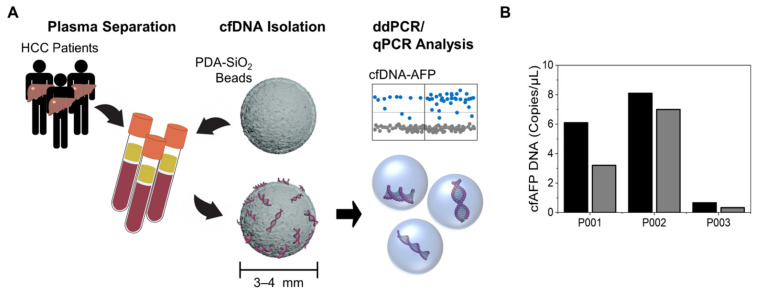
Enhanced detection of cfDNA for its use as a biomarker for the diagnosis and prognosis of HCC. (**A**) A schematic diagram illustrating the cfDNA detection and analysis methods using PDA-SiO_2_ beads. (**B**) The copy numbers of plasma cfAFP-DNA isolated from human plasma samples using either PDA-SiO_2_ beads (black) or QIAamp DNA mini-kit (gray).

**Figure 2 cancers-14-02061-f002:**
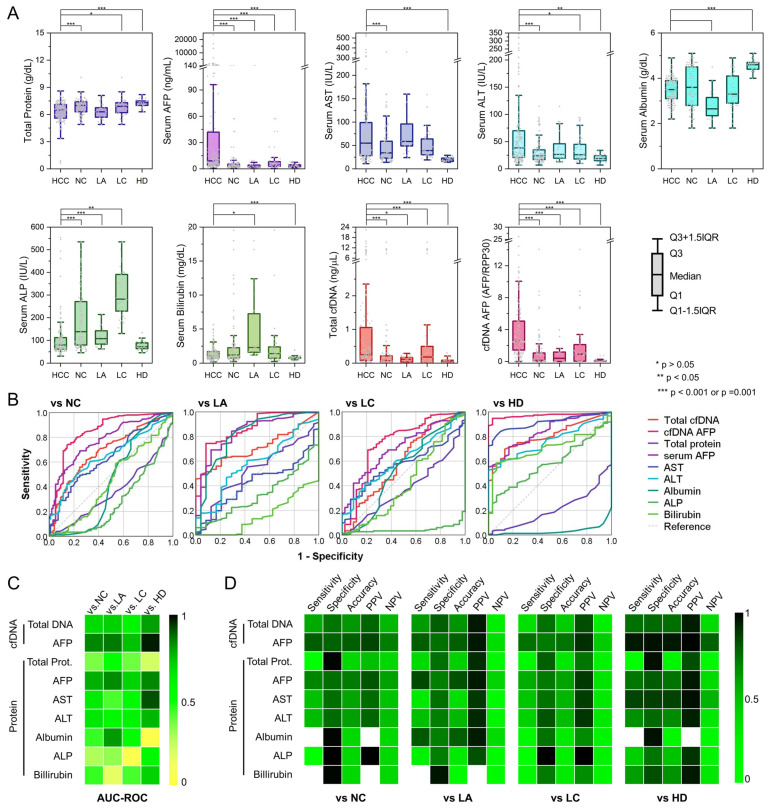
The diagnostic capability of cfDNA for detecting HCC patients from non-cancer (NC) cohorts: (**A**) the expression profiles of serum enzymes, plasma cfDNA, and cfAFP-DNA quantified from a total of 152 HCC patients and 97 non-cancer cohorts, which include 43 patients diagnosed with liver cirrhosis (LC), 24 patients diagnosed with alcoholic liver hepatitis (LA), and 30 healthy donors (HD). (**B**,**C**) ROC curves for diagnosing HCC from NC, LC, LA, and HDs using the expression profiles of serum enzymes, plasma cfDNA, and cfAFP-DNA. (**D**) Diagnostic performance of serum enzymes, plasma cfDNA, and cfAFP-DNA for detecting HCC patients.

**Figure 3 cancers-14-02061-f003:**
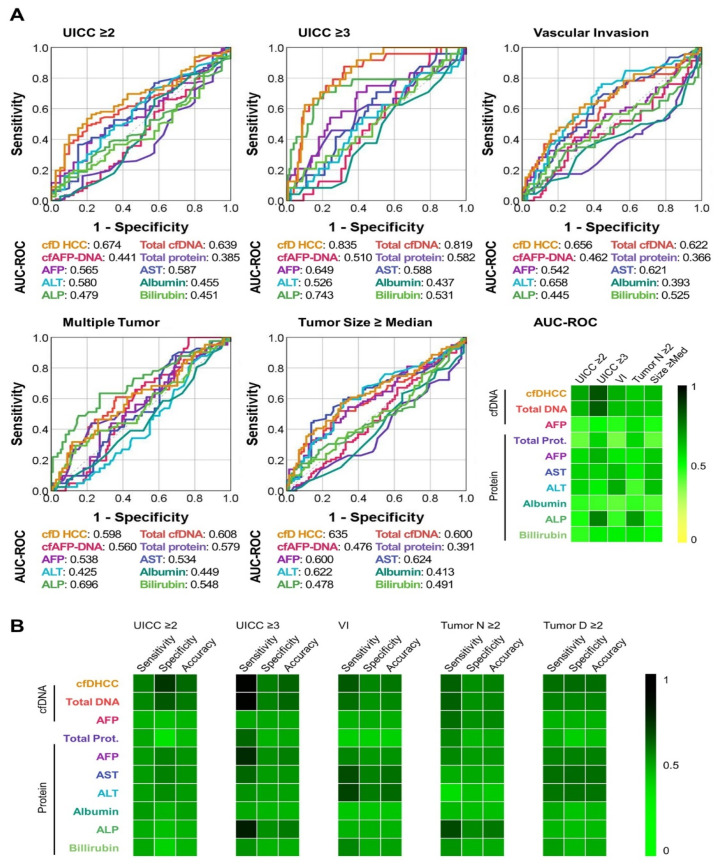
The clinical performance of cfDNA for determining the pathological features of HCC tumors: (**A**) ROC curves for determining the UICC stages, existence of LVI, tumor size, and multifocality of a tumor. (**B**) Sensitivity, specificity, and accuracy of each biomarker for determining the UICC stages, existence of LVI, tumor size, and multifocality of a tumor.

**Figure 4 cancers-14-02061-f004:**
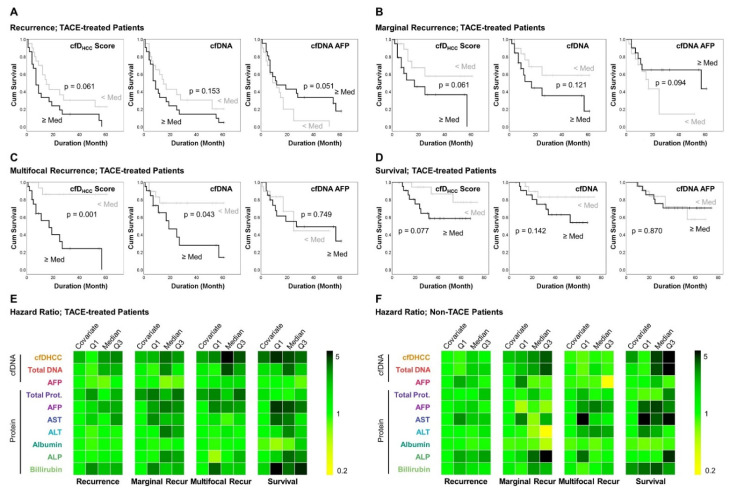
The clinical performance of cfDNA for predicting the survival outcomes: Kaplan–Meier survival analysis for (**A**) recurrence, (**B**) marginal recurrence, (**C**) multifocal recurrence, and (**D**) overall survival of TACE-treated patients. Univariate Cox regression analysis of the serum and plasma biomarkers for (**E**) TACE-treated and (**F**) non-treated patients.

## Data Availability

All data generated from this study are included in this published article and supporting information. Raw data are available from the corresponding author on reasonable request.

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
