# Peer review of "Machine-Learning-Based Clinical Biomarker Using Cell-Free DNA for Hepatocellular Carcinoma (HCC)"

_cancers, 2022, doi:10.3390/cancers14092061_

Round 1

Reviewer 1 Report

Highly Sensitive Circulating Cell-free DNA (cfDNA) Detection in Combination with Machine Learning Algorithm for the Accurate Diagnosis and Prognosis of Hepatocellular Carcinoma (HCC)

The authors presented a machine learning approach to use biomarkers to increase the sensitivity of HCC detection. They tried to inspect the feasibility of ML and PCA technique and cfDNA through liquid biopsy to identify liver cancer.

Some good points about this article:

  • They used model to accomplish this, and their results are interesting
  • The article is well-written and interesting. I personally like this study.
  • Authors managed to modify their article considerably.

Some points that need to be improved before going any further:

Major:

  • Authors stated that: “novel machine-learning algorithm, to provide a platform for improved diagnosis and prognosis of HCC”, where is this novelty as liquid biopsy is already used in many sophisticated articles such as: doi: 10.1186/s12943-019-1043-x, 10.1038/s41598-021-88239-y, 10.3389/fonc.2022.781820, 3390/cancers13040659, and many more.

If authors are doing the same but for liver cancer it needs to be clarified where is your contributions and compare your technical or clinical contributions with these recommended articles.

Methodological data processing here is not clear! Authors provided ample number of results and graphs but there are not explicit and clear discussion about the technical contributions here.

  • This is the case that I would say abundance of information but missing the main part. Contributions, whether clinical or technical need to be explicitly clarified.
  • Moreover, there is a need for clarification regarding the Methodology throughput extracted from regions of interest.
  • The title of this article is three lines?! This needs to be shorten as well.

Author Response

Dear, Reviewer,

Thank you for taking time to review our study with your detailed and thorough comments. We sincerely thank you for your consideration of the many scientific significances of our research. Furthermore, we tried to reflect about your concerns as much as we could. Again, thank you very much for your reviews and comments.

Reviewer 2 Report

The manuscript entitled:" Highly Sensitive Circulating Cell-free DNA (cfDNA) Detection in Combination with Machine Learning Algorithm for the Ac- 
curate Diagnosis and Prognosis of Hepatocellular Carcinoma 
(HCC)" focused on the evaluation of a novel alghoritm able to predict clinical outcome in HCC patients by evaluating cfDNA is well written and requires minor modifications to be accepted for the publication on this journal:

  • In the material and method section, please, could the authors describe in details pre analytical steps for the managing of liquid biopsy specimens? In my opinion, collection time, plasma separation modalities may represent an additional point for this manuscript.
  • In the results section, please, could the authors also demonstrate if a statistically relevant correlation may be observed between pathological features (staging, grading) and molecular data between cancer and non cancer (patients with inflammatory pathology) patients? 
  • In the supplementary section, pelase, could the authors report DNA quantification data of cfDNA for enrolled patients? In my opinion, this aspect may represent a point of strenght for this manuscript

Author Response

(The authors gave the same response as above.)

Round 2

Reviewer 1 Report

Authors managed to respond my comments well. I don't have any further questions.